# Observation of ¹H-¹H J-couplings in fast magic-angle-spinning solid-state NMR spectroscopy

Daria Torodii [1], Jacob B. Holmes [1], Kristof Grohe[2], Rodrigo de Oliveira-Silva [3], Sebastian Wegner[2], Dimitrios Sakellariou [3] ✉ & Lyndon Emsley [1] ✉

While ¹H-¹H J-couplings are the cornerstone of all spectral assignment methods in solution-state NMR, they are yet to be observed in solids. Here we observe ¹H-¹H J-couplings in plastic crystals of (1S)-(−)-camphor in solid-state NMR at magic angle spinning (MAS) rates of 100 kHz and above. This is enabled in this special case because the intrinsic coherence lifetimes at fast MAS rates become longer than the inverse of the ¹H-¹H J couplings. For example, at 160 kHz MAS the coherence lifetimes are longer than 20 ms, corresponding to refocused linewidths of less than 15 Hz. As a result, we are able to record two-dimensional ¹H-¹H J resolved spectra that allow the observation and measurement of ¹H-¹H J-couplings in solid camphor. The J-couplings also lead to unambiguous through-bond correlations in ¹H-¹H refocused incredible natural abundance double quantum transfer (INADEQUATE) and uniform-sign cross-peak double-quantum-filtered correlation spectroscopy (UC2QFCOSY) experiments.

¹H-¹H J couplings were first observed in 1951 by Hahn and Maxwell[1]. Since then they have become ubiquitous as detailed structural probes in all areas of chemistry, with the analysis of multiplet patterns forming the basis of modern structural analysis by NMR[2,3], and the seminal Karplus relations providing a direct link between $^3J_{HH}$ and dihedral angles[4,5]. However, more than 70 years after their first observation, ¹H-¹H J-couplings are yet to be observed in solids.

There are two main barriers to the observation of ¹H-¹H J-couplings in solids. The first is that ¹H-¹H J-couplings are relatively small, with magnitudes typically less than 20 Hz. The second is that the linewidths in solid-state ¹H NMR spectra of natural abundance protonated solids are typically much broader than this. Indeed, the main contribution to the linewidth in static samples is the homonuclear dipolar coupling, which typically leads to linewidths of around 40 kHz in organic solids. A great deal of effort has been focused over the last 65 years on removing dipolar broadening by magic angle spinning (MAS)[6,7] and by the application of radio-frequency pulse sequences[8,9]. The best ¹H resolution today is achieved with MAS at rates around 100 kHz, where in favorable cases linewidths of 100–300 Hz can be achieved[10–15].

Recently, it was shown that ¹H linewidths of ~120 Hz could be achieved for the $CH_2$ group of O-phospho-L-serine at 160 kHz MAS at 1200 MHz[16]. These linewidths are nevertheless still an order of magnitude larger than ¹H-¹H J-couplings.

Homonuclear J-couplings between weakly coupled spins (e.g. $^{31}P$-$^{31}P$, $^{13}C$-$^{13}C$, $^{15}N$-$^{15}N$, $^{29}Si$-$^{29}Si$, $^{11}B$-$^{11}B$, etc.) have been observed previously in solids[17–21]. Heteronuclear J-couplings between weakly coupled nuclei and protons have also been observed[22], though these experiments are much more challenging. In all these cases above, where quantitative measurements of J-couplings were possible, they have been related to important structural features, such as for example hydrogen bonding[18] or agostic interactions[22] and serve as a motivation to introduce more J-based correlation methods for solids[23].

In all these cases, the linewidths in the spectrum were also typically significantly larger than the J-couplings. However, because the resonances were inhomogeneously broadened, it was found that the coherence lifetime ($T_2'$) in a spin-echo experiment can actually be much longer than the inverse of the J-coupling[24,25]. Thus, J-couplings

¹Institut des Sciences et Ingénierie Chimiques, École Polytechnique Fédérale de Lausanne (EPFL), CH-1015 Lausanne, Switzerland. ²Bruker BioSpin GmbH & Co KG, 76275 Ettlingen, Germany. ³KU Leuven, M2S, cMACS, Celestijnenlaan 200F, 3001 Leuven, Belgium. ✉e-mail: dimitrios.sakellariou@kuleuven.be; lyndon.emsley@epfl.ch

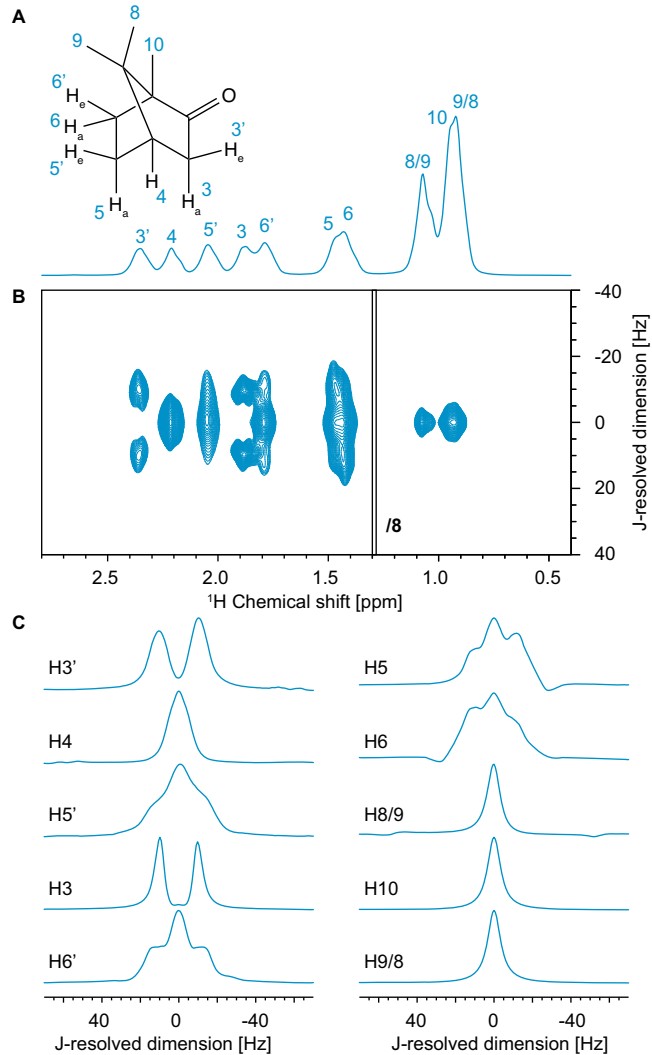

**Fig. 1 | 2D JRES spectrum of camphor. A** Molecular structure of camphor with labels for each proton site and the 800 MHz one-dimensional ¹H spectrum acquired at 169.880 kHz MAS using a rotor-synchronized spin echo sequence for background suppression. The *e* and *a* subscripts represent equatorial and axial proton positions, respectively. **B** Contour plot of the 800 MHz two-dimensional ¹H-¹H JRES spectrum of camphor acquired at 168.571 kHz MAS, at 298 K with an FID resolution of 2.8 Hz. The contour levels were decreased by a factor 8 in the 0.4–1.3 ppm region. **C** Columns parallel to $\omega_1$ extracted from the spectrum of (**B**) at each of the isotropic shifts in $\omega_2$. The columns are normalized to the same maximum intensity. In (**B**) and (**C**) only the center region, between ±40 Hz and ±70 Hz in $\omega_1$, respectively, is shown. The full spectrum is shown in Fig. S3. (Pulse sequences and acquisition parameters for all the spectra are given in the Supplementary Table 2. The link to the raw data is given in the SI).

could be measured or used to generate correlations in spin-echo based experiments[18,22,25].

In this respect, it has recently been shown that in the fast MAS regime, the residual dipolar broadening of ¹H resonances is a mixture of non-refocusable and refocusable contributions, in roughly equal parts[10,14,15,26,27]. Furthermore, a detailed analysis shows that in addition to the residual dipolar broadening, the one-dimensional linewidth contains contributions from anisotropic bulk magnetic susceptibility (ABMS), and inherent structural disorder[10,15,28]. Both of these latter contributions are inhomogeneous (excluding the effects of potential higher order cross terms), and thus refocusable.

We thus hypothesize that ¹H linewidths at MAS rates above 100 kHz may be entering a regime similar to carbon-13 linewidths at slower MAS rates, and that J-couplings may become observable in spin-

echo based experiments. For example, two-dimensional J-resolved spectroscopy (2D JRES) is the reference experiment to measure homonuclear J-couplings in solution NMR[3,29,30], since it is based on a spin-echo which removes all refocusable interactions (including the chemical shift) in the indirect dimension. While this is still unlikely to be the case for rigid molecular solids, we further hypothesize that the mobility present in plastic crystals may attenuate the dipolar network sufficiently so that in combination with the fastest MAS rates achievable today ¹H-¹H J-couplings might become observable in some special cases.

One of such cases is camphor, having a plastic phase from 242 to 374 K[31], in which the individual molecules undergo rapid isotropic reorientation around their crystallographic centers of gravity. This weakens the ¹H-¹H dipolar coupling network by averaging out intramolecular dipolar couplings but leaving inter-molecular couplings intact. This results in linewidths in the one-dimensional ¹H spectrum of Fig. 1A at 170 kHz MAS of between around 60 and 70 Hz for the resolved peaks.

Here we show that at MAS rates of 170 kHz, ¹H coherence lifetimes in (1S)-(−)-camphor (Fig. 1A) can lead to refocused linewidths less than 10 Hz at natural ¹H abundance. This allows us to observe resolved homonuclear ¹H-¹H J-couplings for the first time in a solid. We do this using a phase sensitive 2D JRES experiment[22,29], where we resolve and also accurately measure homonuclear J-couplings. We then demonstrate that the J-couplings in (1S)-(−)-camphor lead to unambiguous through-bond correlations in ¹H-¹H refocused INADEQUATE and UC2QFCOSY experiments.

## Results and discussion

Fig. 1B shows an 800 MHz phase-sensitive ¹H-¹H 2D JRES spectrum of camphor obtained at 168.571 kHz MAS using a Bruker 0.4 mm CPMAS HCN probe. In the 2D JRES spectrum of Fig. 1B we clearly observe well resolved splittings in the indirect ($\omega_1$) dimension for several of the resonances. Fig. 1C shows the columns taken along $\omega_1$ at each of the peak positions in $\omega_2$. The J-coupling between H3 and H3' causes a splitting of both signals into clear doublets in $\omega_1$ (this is the largest J-coupling, at 18.1 Hz, observed in the solution-state NMR measurements shown in Table S4). Notably, the linewidths of each of the four peaks in the two doublets are 9 and 5 Hz, respectively. The methyl group protons labeled 8, 9 and 10 show narrow singlets in $\omega_1$ with linewidths of around 8 Hz. From solution NMR, the methyl groups are expected to only have very weak J-couplings ( < 0.5 Hz), notably: H8 with H3', H4 and H6; H9 with H3 and H4; H10 with H3, H4, H5 and H6'. We thus conclude that the nonrefocusable ¹H linewidths observed for camphor at 170 kHz MAS are between 5 and 10 Hz. (We note for example that H3' is expected to have additional couplings of around 5 Hz to H4, and 3 Hz with H5'. The columns extracted from the ¹H 2D JRES spectrum acquired at 170 kHz MAS overlaid with a simulated spectra where all the peaks have a linewidth of 5 Hz and subject to all the J-splittings measured in solution NMR is shown in Fig. S2).

The resonances corresponding to H4, H5, H5', H6, and H6' are all visibly broadened by J-couplings in $\omega_1$. H6, H6', and H5 clearly appear as triplet like structures, with a (roughly) 1:2:1 intensity pattern. Indeed, in solution-state, these three protons all experience two similar J-couplings of 11 to 12 Hz (Table S4) and a third coupling of around 3 - 4 Hz. We expect this to lead to the slightly broadened pseudo-triplet structure we observe. H5', on the other hand, has two J-couplings of 12.0 and 11.3 Hz, but also three smaller J-couplings of 4.5, 4.0, and 3.2 Hz in solution, resulting in a complex multiplet lineshape, leading to clear broadening in the spectrum of Fig. 1B, but where the fine structure is unresolvable in the solid-state.

We have measured ¹H-¹H 2D JRES spectra for MAS rates from 100 kHz to 170 kHz, and the extracted columns at the different rates for H3 and H3' are shown in Fig. 2A, (and for all the peaks in Fig. S4.)

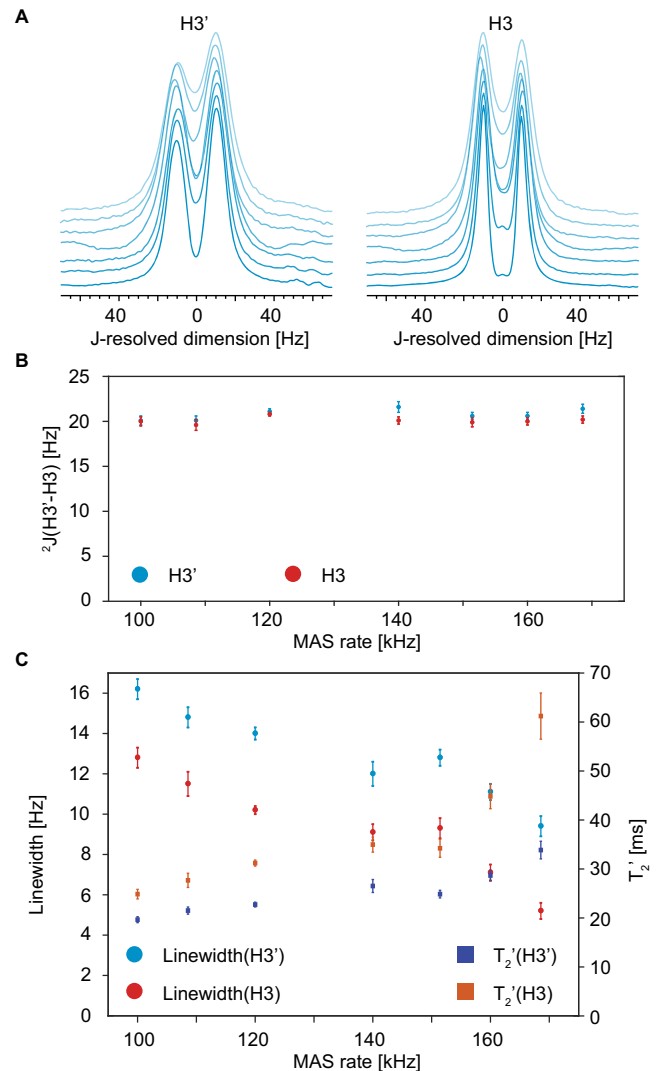

**Fig. 2 | MAS rate dependence of lineshapes and J-couplings. A** Columns parallel to $\omega_1$ extracted from 800 MHz two-dimensional $^1$H-$^1$H JRES spectra at the isotropic shifts of H3 and H3' in $\omega_2$ obtained at MAS rates between 100 kHz (upper) and 169 kHz (lower) at sample temperature of 295 K with an FID resolution of 2.8 Hz. **B** The value of the J-coupling obtained by fitting the columns corresponding to H3' and H3 as a function of MAS rate in blue and red, respectively. **C** Fitted linewidths as a function of MAS rate labeled with blue and red circles and coherence lifetimes labeled with purple and orange squares respectively for H3' and H3. The datapoints shown in (**A**), (**B**) and (**C**) were obtained at MAS rates of 100, 109, 120, 140, 151, 160 and 170 kHz. Details of the linewidths fitting procedure and a table of the fitted values for the linewidths are given in Section 3 of the SI. The error bars in (**B**) and (**C**) are the overall errors on the fits obtained by combining the errors from each fitted parameter, as detailed in the "Overall Parameter Error" section of the Mathematica notebook available in the SI.

Resolution is clearly best at the fastest MAS rate, but interestingly the doublets are already visible for H3 and H3' at 100 kHz MAS.

Figure 2C shows the evolution of the fitted linewidth of H3 and H3' in $\omega_1$ as a function of MAS rate. As expected, the linewidths decrease significantly with increasing MAS rate. At 100 kHz MAS, the linewidths are 12.8 ± 0.5 and 16.2 ± 0.5 Hz, respectively. This is 3-5 times narrower than the 1D linewidths of the same peaks obtained in the 1D MAS experiment (Fig. S8) using the same fitting model, which are 54 ± 2 Hz and 72 ± 6 Hz, respectively. Furthermore, at 169 kHz MAS the H3 and H3' columns narrow down to 5.2 and 9.4 Hz, respectively, nearly a 2-fold decrease compared to 100 kHz MAS. The exact form of the linewidth dependence on MAS rate cannot be determined reliably, since in the prototype 0.4 mm MAS probe used here the magic angle needed to be readjusted at each spinning rate, introducing some degree of error. Notably, this explains the apparent increase in the refocused linewidths at 150 kHz MAS for H3' and H3, which is within the fitting error of ±0.5 Hz. Quantitative studies and comparison with detailed theoretical predictions[26,27] will be the subject of future investigation. However, the refocused linewidths do not plateau at the fastest MAS rates, and it is assumed that even narrower lines will be obtained in 2D JRES spectra at faster MAS rates.

It should be noted that all the solid-state 2D JRES spectra acquired here contain weak folded artifacts in the $\omega_1$ dimension (Fig. S3), that can be found in any 2D JRES spectrum[3] (for liquids or solids) and which are due here to imperfect 180° pulses in combination with the use of an incomplete phase cycle. (We used 4 steps instead of 16 to save time in these experiments which require very high digitization in $t_1$.) These artifacts can be removed, as shown in Fig. S7 with a spectrum acquired using a simpler pulse sequence that leads to broader mixed-phase lineshapes in F1, but where the artifacts are absent. The spectral width in $\omega_1$ was therefore adjusted to minimize overlap with the center peaks. Nonetheless, partial overlap with the center peaks lead to slight asymmetry in the multiplet patterns (Fig. S3)

Given that the $^1$H-$^1$H J couplings are clearly visible in the 2D JRES spectra, we suggest that it should be possible to record two-dimensional J-based through-bond correlation spectra for (1S)-(−)-camphor by using pulse sequences that produce in-phase correlations using transfer schemes that refocus inhomogeneous broadening. To this end, Fig. 3 shows the 900 MHz $^1$H-$^1$H refocused INADEQUATE[25,32–34] (A) and UC2QFCOSY[35] (B) spectra of camphor recorded at 100 kHz MAS using an echo delay equal to one thousand rotor periods (optimal for the evolution of 25 Hz J-coupling).

All the correlations seen in both spectra correspond to all the expected pairs of through-bond J-coupling partners identified from solution, except for the weak J-couplings $^4$J(H3'-H5') and $^3$J(H5'-H6) (expected at around of 3.2 and 4.0 Hz from solution). In addition, the solution-state DQF-COSY spectrum (Fig. S9) shows weak correlation peaks between H4 and H8, H9, and H10, although their J-coupling constants were too small to be quantified. Interestingly, the J-based correlations between H4 and H8/9, H10, and H9/8 are also observed in the solid-state refocused INADEQUATE and UC2QFCOSY spectra. Most importantly, we see no correlations in these spectra that can be ascribed to through-space, dipolar driven, transfer.

In the refocused INADEQUATE spectrum in Fig. 3A we notice some negative peaks at ($\omega_1$, $\omega_2$) frequencies corresponding to (H5', H6' + H6), (H6', H5' + H5) and (H5, H5' + H6'). These are examples of relayed peaks between two nuclei that share a common J-coupling partner and are typically observed in INADEQUATE spectra of abundant nuclei in solution. Similarly, some of the correlation peaks (e.g. H3', H4) show a mixed in-phase / anti-phase character, which is also due to three-spin effects. Both of these effects have been previously reported in the solid-state in $^{13}$C-$^{13}$C INADEQUATE spectra of $^{13}$C isotopically enriched samples[36,37].

We note that there is an example of the $^1$H-$^1$H refocused INADEQUATE pulse sequence being previously used to generate through space correlation in rigid solids at ~60 kHz MAS[38], where double quantum coherences were generated by the evolution of a three-spin term in the homogeneous homonuclear dipolar Hamiltonian. In the case of camphor at >100 kHz MAS, the dipolar coupling network is significantly weaker than in rigid solids, and this three-spin dipolar mechanism is not active[38]. Indeed, dipolar mechanisms in plastic

The H3-H3' scalar coupling was then measured quantitatively by fitting the extracted columns for H3 and H3' at each MAS rate. (Fig. 2B) The J-splitting is measured to be 20.3 ± 0.8 Hz. This can be compared to the value in a DMSO-d$_6$ solution of camphor measured to be 18.1 ± 0.3 Hz. (Figs. S5 and S6) The fitting procedure is described in detail in the SI.

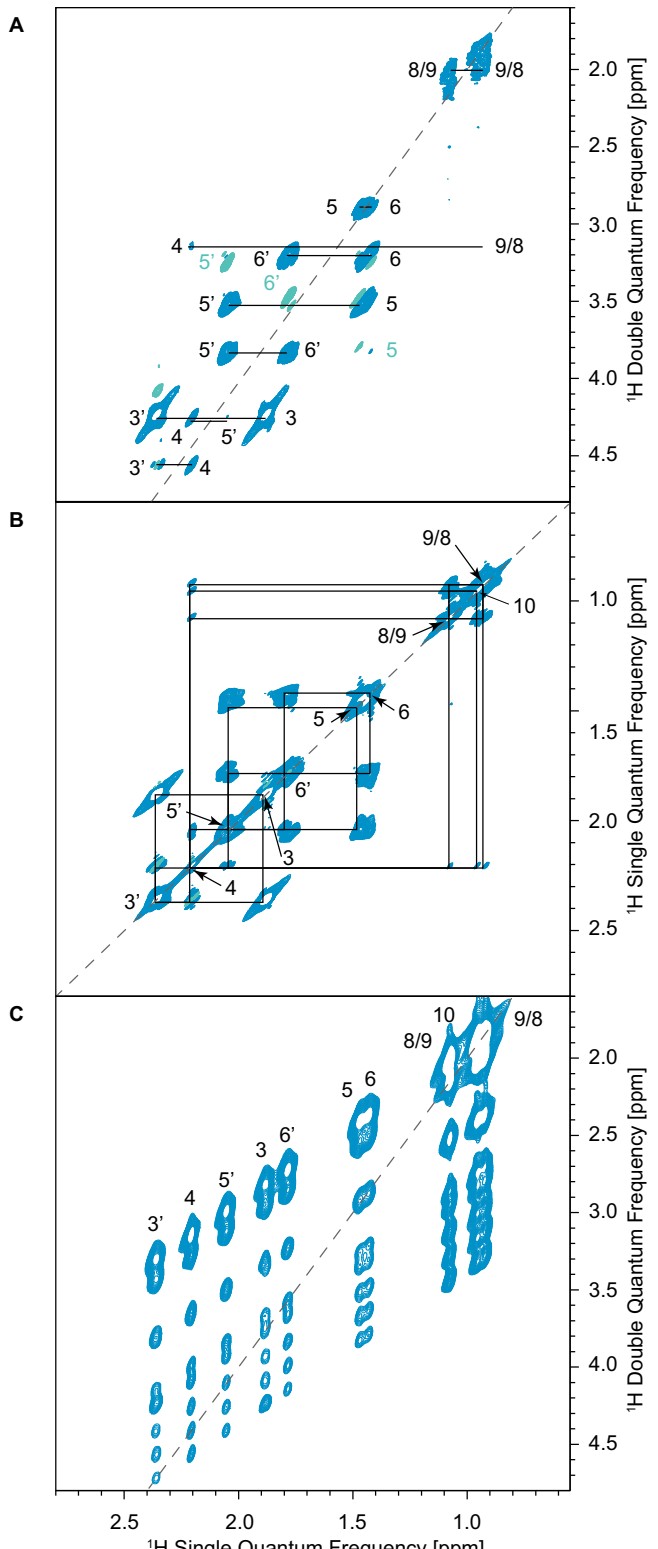

**Fig. 3 | Contour plots of 900 MHz 2D spectra of microcrystalline camphor.**
**A** $^1$H-$^1$H refocused INADEQUATE with 20 ms 2τ delays for excitation and reconversion of double-quantum coherence, **B** UC2QFCOSY with 20 ms 2τ delays, and
**C** BABA-xy16 with 80 μs excitation and reconversion periods. All three spectra were acquired at 100 kHz MAS at room temperature. Positive and negative contours are shown in blue and green, respectively. In (**A**), the labels in green indicate relayed peaks. The dotted grey lines indicate the spectral 2:1 and 1:1 "diagonals." The solid black lines indicate the connectivities between cross peaks. (Pulse sequences and acquisition parameters for all the spectra are given in the Supplementary Table 3. The link to the raw data is given in the SI.)

crystals would yield correlations between all pairs of proton sites. This is due to the correlations being driven exclusively by inter-molecular dipolar couplings, since the intra-molecular couplings are averaged to zero by molecular motion. The average values of the inter-molecular $^1$H-$^1$H couplings are rather similar. (The estimated distributions of the averaged $^1$H-$^1$H dipolar couplings for camphor are given in the SI.)

In very stark contrast to the refocused INADEQUATE and UC2QFCOSY spectra, the dipolar-recoupled $^1$H-$^1$H BABA-xy16 spectrum shown in Fig. 3C does indeed yield correlations between all pairs of proton sites with comparable intensities, as expected.

This demonstrates unambiguously that the scalar $^1$H-$^1$H J-couplings are the only interactions that lead to the correlations in the 100 kHz MAS $^1$H-$^1$H refocused INADEQUATE and UC2QFCOSY spectra of camphor.

In conclusion, we have observed and measured $^1$H-$^1$H J-couplings in solid (1S)-( − )-camphor at MAS rates of 100 kHz and above. This was enabled in this case because the intrinsic refocused linewidths (proportional to the inverse of the transverse dephasing time $T_2$') at fast MAS become smaller than the $^1$H-$^1$H J couplings. The fast molecular dynamics present in camphor lead to a weakened dipolar network, which is in contrast to the refocused linewidths measured for rigid organic solids (*e.g.* tyrosine) that are currently still slightly larger (typically 50-200 Hz at 100 kHz MAS)[15] than typical $^1$H-$^1$H J-couplings and which prevents the approach here from being broadly applicable to rigid solids.

Using camphor, we were therefore able to record two-dimensional $^1$H-$^1$H JRES spectra that allow us to observe and quantitatively measure $^1$H-$^1$H J-couplings, and this even though the one-dimensional spectral linewidths are here roughly 3 times broader than the J-couplings.

Furthermore, this also enabled the acquisition of two-dimensional $^1$H-$^1$H J-mediated through-bond correlation experiments for camphor, exemplified with refocused INADEQUATE and UC2QFCOSY spectra, that show exclusively J-mediated cross peaks.

## Methods
(1S)-(−)-camphor was purchased from Alfa Aesar and used without further recrystallization. The powder was packed in 0.7- and 0.4-mm rotors after being crushed with a mortar and pestle.

### NMR experiments
All the 1D and 2D JRES spectra between 100 and 169.880 kHz MAS were acquired on an 18.8 T Bruker Avance Neo spectrometer corresponding to a $^1$H frequency of 800 MHz using a Bruker 0.4 mm HCN CP-MAS probe. The sample temperature was regulated to 295 K using VT flow. At each MAS rate, after temperature stabilization, the magic-angle was reset by maximizing signal intensity in a 1D $^1$H experiment using a spin-echo sequence with echo delays equal to 15 ms. The spinning was controlled by Bruker MAS III unit. The MAS instability was estimated to be roughly ± 100 Hz at all MAS rates on the 0.4 mm probe.

The refocused INADEQUATE, UC2QFCOSY and BABA-xy16 spectra at 100 kHz MAS were acquired on a 21.14 T Bruker Avance Neo spectrometer corresponding to a $^1$H frequency of 900 MHz using a Bruker 0.7 mm room temperature HCN CP-MAS probe. The temperature was regulated to 295 K using a VT flow at 285 K. The magic angle was optimized directly on the sample by maximizing the T2'. The spinning was controlled by Bruker MAS III unit. The MAS instability at 100 kHz MAS on the 0.7 mm probe was estimated to be roughly ± 100 Hz.

A States-TPPI acquisition scheme was used in all 2D experiments to obtain phase-sensitive two-dimensional spectra. All spectra were phase and baseline corrected. No window functions were applied prior to Fourier transformation.

### Fitting results
The fitting of the 2D JRES columns H3 and H3' acquired at 100–168 kHz MAS was carried out in Wolfram Mathematica 13.2 student edition.

Each data set was fit to a Lorentzian function with the J-coupling constant, linewidth and amplitude used as fitting parameters. The Mathematica notebook called Camphor_Solids_LW_clean.nb and the fitting results are given in the SI.

## Supporting information
Experimental and fitting details, additional figures and tables, a Mathematica notebook, and a link to all the raw NMR data.

## Data availability
The NMR raw data are available from the Zenodo repository [https://doi.org/10.5281/zenodo.14186567] in JCAMP-DX version 6.0 standard format and the original TopSpin format. All data and scripts are available under the license CC-BY-4.0 (Creative Commons Attribution-ShareAlike 4.0 International https://creativecommons.org/licenses/by-sa/4.0/).

## Code availability
The Mathematica notebooks used for fitting the 2D JRES columns and to calculate the dipolar networks are available in the supporting information and/or from the Zenodo repository [https://doi.org/10.5281/zenodo.14186567]. All data and scripts are available under the license CC-BY-4.0 (Creative Commons Attribution-ShareAlike 4.0 International https://creativecommons.org/licenses/by-sa/4.0/).

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

## Acknowledgements
We thank Dr. Federico De Biasi for assistance with the experiments. This work has been supported by the Swiss National Science Foundation Grant No. 200020_212046 (L.E.), the European Union's Horizon 2020 research and innovation programme under Grant Agreement No. 101008500 (PANACEA) (L.E.), and by the KU Leuven Grant STG-18-00289 (D.S.).

## Author contributions
Conceptualization: D.S., L.E. and D.T. Methodology: D.T., D.S. and L.E. Investigation: D.T., J.B.H., K.G., S.W. R. O. S., Visualization: D.T., J.B.H. and L.E. Supervision: L.E. Writing (original draft): D.T. and writing (review and editing): D.T., J.B.H., K.G., S.W., R. O. S., D.S. and L.E.

## Competing interests
The authors declare no competing interests.
