## [Transparent Peer Review file · Nature Communications]

Observation of ^1H - ^1H J-Couplings in Fast Magic-Angle-Spinning Solid-State NMR Spectroscopy

Corresponding Author: Professor Lyndon Emsley

Version 0:

Reviewer comments:

Reviewer #1

(Remarks to the Author)

The paper presents the first observation of homonuclear ^1H - ^1H J coupling in solids. Despite the importance of ^1H - ^1H through-bond J coupling in solution NMR for understanding inter-proton connectivities, it has never been observed in solids. This is due to the ^1H - ^1H homonuclear dipolar interaction in solids, which results in non-refocusable linewidth broader than those of ^1H - ^1H J coupling, thus preventing its observation. In the current study, the authors employed a very fast MAS frequency of 170 kHz to minimize non-refocusable linewidth and successfully observed ^1H - ^1H J coupling in a plastic crystal. This development opens a new avenue for structural analysis of solids, akin to solution NMR, although the method is currently applicable only to plastic crystals. The paper is well written and all the experimental results together with the discussion support the conclusions. The topic is of broad interest, as ^1H NMR and analysis of ^1H - ^1H J coupling are fundamental steps in structure analysis for organic chemists. I recommend publication in Nature Communications after addressing the following minor points:

- 1) Additional explanation would benefit non-NMR experts in understanding the details. For example, the comparison between J-based and dipolar-based DQ/SQ correlations is unclear. In camphor, all intramolecular dipolar couplings are averaged out due to isotropic tumbling motion in plastic crystals, so ^1H - ^1H dipolar coupling appears only between intermolecular ^1H pairs. This results in ^1H - ^1H couplings for all possible pairs with the same intensities, thus, all possible ^1H s pairs exhibit the dipolar-driven DQ/SQ correlations. While the authors tried to explain this in the last sentence of the left column of page 4, it remains unclear for non-experts.
 - 2) Refocused INADEQUATE has already been applied to extract ^1H - ^1H coupling information in rigid solids, utilizing residual dipolar coupling rather than ^1H - ^1H J coupling (Phys. Chem. Chem. Phys. 13 (2011) 8024-8030). This article should be cited, as it supports the absence of dipolar-driven cross peaks in Fig 3a. Indeed, dipolar-driven correlations should appear between all ^1H s pairs as explained above.
 - 3) Remove the period line 6 of the left column of page 1. “3JHH.” Should be “3JHH”
- These revisions will enhance the clarity and completeness of the paper for a wider audience.

Reviewer #2

(Remarks to the Author)

This is the first solid-state NMR observation that resolves ^1H - ^1H J couplings. It is made possible by the highest magic-angle spinning available today (0.4mm and 160kHz) and the sample used. Camphor is among a few molecular crystals that have much reduced proton homonuclear dipolar couplings by the isotropic molecular rotation. Only the intermolecular dipolar couplings are present. Even with this, the T_2' has not reached a plateau with the spinning speed. Therefore I feel pessimistic about the impact and outlooks applied to other more 'normal' molecules and solid samples.

The 2D correlation with J and D couplings turned out expected. There are still some visible features or 'artifacts' that are likely caused by residual homonuclear dipolar couplings. For camphor, various proton sites are well resolved. Do the shift differences impact the MAS average of the homonuclear dipolar couplings? Have the authors tried the other 'well known' molecular crystal adamantane in which the CH and CH₂ sites have nearly identical shift. I would be curious on the effect of shift degeneracy on the MAS averaging of dipolar couplings. The setting of magic-angle is very critical for observing small J-couplings and it will become even more for molecules with stronger ^1H homonuclear dipolar couplings. I would be keen to

see how the J-res spectra change with the angle setting instead just making the T2' longer stated for the angle calibration. Also, the low-contour plot for the 2D J-res in SI shows spinning sidebands in F1. Would that problem be solved by rotor-sync t1 evolution time?

Reviewer #3

(Remarks to the Author)

This study explores the observation and use of 1H-1H J-couplings in plastic crystals using fast MAS solid-state NMR spectroscopy. The results are very interesting, especially the use of a cutting-edge 0.4 mm probe to enable the fastest spinning at around 170 kHz. Nevertheless, I have some major concerns before consideration for publication.

1. Figure 1c, why the height of doublet is not equal, such as H3' and H3?
2. Left top of Page 4, please explain why there are "folded artifacts in the w1 dimension (Figure S3)". This should be briefly mentioned in the main text instead of simply citing other works. This is very important, because it is related to the correct setting of experimental parameters.
3. In Table S3, the spectral width of F1 dimension for BABA-xy16 experiment is 9999.966 kHz? Please double check.
4. Figure 2c, there is an abnormal increase of linewidth at 150 kHz. Could the authors explain why?
5. The authors mentioned that coherence lifetimes are longer than 20 ms at 160 kHz in the abstract. Please provide the experimental data about coherence lifetimes at different spinning rate.
6. Figure S7, it seems that the proton linewidths do not decrease with increasing spinning rate. Could the authors comment on this strange behavior?
7. Figure S6, it is a solution JRES spectrum for camphor dissolved in DMSO. Why does the spectrum look so noisy, especially at the chemical shift of 1.3ppm? Usually there should be a clear splitting for each peak in solution NMR spectrum.
8. Usually the JRES experiment is amplitude modulated, but the authors used phase sensitive JRES experiment. Please explain the advantages of phase sensitive experiment and cite the relevant literatures about JRES experiment.
9. Caption of Figure 2, please use "JRES" instead of "J-RES" for consistency through the main text.
10. Figure 3A, for the correlation between H3' and H4, why the peak is negative on the left but positive on the right of the diagonal peak? Similar negative peaks are observed in Figure 3B for the (H3', H4) cross peaks. Besides, why there are strong t1-noise at 1.0ppm in Figure 3A, but it is not observable in Figure 3B.
11. One of the key issues for observation of 1H-1H J couplings in solid-state NMR is to make sure that the proton dipolar couplings are completely suppressed. In fact, the strength of residual 1H-1H dipolar couplings could be on the same level as J couplings. So, how do the authors can guarantee that the dipolar couplings are completely removed, and they are actually observing the 1H-1H J coupling instead of residual 1H-1H dipolar coupling. Particularly, the authors mentioned that the "magic angle needed to be readjusted at each spinning rate". Insufficiently accurate magic angle may also introduce additional residual dipolar couplings. Careful discussion about this point is very important, since we can see that the obtained J(H3'-H3) is larger than that obtained from solution NMR.
12. The authors mentioned in the main text "a detailed analysis shows that in addition to the residual dipolar broadening, the one-dimensional linewidth contains contributions from anisotropic bulk magnetic susceptibility (ABMS), and structural disorder. Both of these latter contributions are inhomogeneous, and thus refocusable." The statement may not be accurate, since I believe the higher-order ABMS component is actually not refocusable. Similarly, the inherent structural disorder induced chemical shift dispersion can be refocusable?
13. In the main text, the authors should summarize all the obtained J values by fitting the columns in Figure 1c into a Table in order to compare the experimental results to that of solution NMR.
14. Please provide important experimental parameters in the Figure captions. For example, for Figure 3, the recoupling time for each experiment should be provided.

Version 1:

Reviewer comments:

Reviewer #2

(Remarks to the Author)

The authors responded adequately with revisions to my comments. I would recommend for publication without further revision

Reviewer #3

(Remarks to the Author)

I appreciate the authors' great effort in improving the quality of the manuscript. Before the acceptance, I still have some minor comments.

1. Regarding to the "weak folded artifacts in the w1 dimension" in Figure 3a (or Figure S3), can the authors comment on how does the folding happen in the w1 dimension and is there any approach for overcoming this problem?
2. In Table S3, the author corrected "the spectral width of F1 dimension for BABA-xy16 experiment" to be 10000 kHz, i.e. 10 MHz. Why using such a huge spectral width? Such spectral width is also completely not rotor-synchronized. Why?
3. In the revised SI, somehow Table S4 is missing, but TableS1-S3 and Table S6-S8 are there.

Version 2:

Reviewer comments:

Reviewer #3

(Remarks to the Author)

The authors carefully address all those comments. I am glad to recommend the publication of this study without further revision.

REVIEWER COMMENTS

Reviewer #1 (Remarks to the Author):

The paper presents the first observation of homonuclear 1H - 1H J coupling in solids. Despite the importance of 1H - 1H through-bond J coupling in solution NMR for understanding inter-proton connectivities, it has never been observed in solids. This is due to the 1H - 1H homonuclear dipolar interaction in solids, which results in non-refocusable linewidth broader than those of 1H - 1H J coupling, thus preventing its observation. In the current study, the authors employed a very fast MAS frequency of 170 kHz to minimize non-refocusable linewidth and successfully observed 1H - 1H J coupling in a plastic crystal. This development opens a new avenue for structural analysis of solids, akin to solution NMR, although the method is currently applicable only to plastic crystals.

The paper is well written and all the experimental results together with the discussion support the conclusions. The topic is of broad interest, as 1H NMR and analysis of 1H - 1H J coupling are fundamental steps in structure analysis for organic chemists.

We thank the reviewer for their positive appreciation.

I recommend publication in Nature Communications after addressing the following minor points:

1) Additional explanation would benefit non-NMR experts in understanding the details. For example, the comparison between J-based and dipolar-based DQ/SQ correlations is unclear. In camphor, all intramolecular dipolar couplings are averaged out due to isotropic tumbling motion in plastic crystals, so 1H - 1H dipolar coupling appears only between intermolecular 1H pairs. This results in 1H - 1H couplings for all possible pairs with the same intensities, thus, all possible 1H s pairs exhibit the dipolar-driven DQ/SQ correlations. While the authors tried to explain this in the last sentence of the left column of page 4, it remains unclear for non-experts.

The sentence has been rephrased to make this clear to the non-expert.

2) Refocused INADEQUATE has already been applied to extract 1H - 1H coupling information in rigid solids, utilizing residual dipolar coupling rather than 1H - 1H J coupling (Phys. Chem. Chem. Phys. 13 (2011) 8024-8030). This article should be cited, as it supports the absence of dipolar-driven cross peaks in Fig 3a. Indeed, dipolar-driven correlations should appear between all 1H s pairs as explained above.

Indeed, many approaches have been used to obtain 1H - 1H dipolar information from double-quantum / single-quantum experiments (including the BABA type experiments used here). We have now included a reference to the paper mentioned by the reviewer, as suggested, to support the absence of dipolar driven cross peaks.

3) Remove the period line 6 of the left column of page 1. "3JHH." Should be "3JHH"
These revisions will enhance the clarity and completeness of the paper for a wider audience.

Corrected.

Reviewer #2 (Remarks to the Author):

This is the first solid-state NMR observation that resolves 1H - 1H J couplings. It is made possible by the highest magic-angle spinning available today (0.4mm and 160kHz) and the sample used. Camphor is among a few molecular crystals that have much reduced proton homonuclear dipolar couplings by the isotropic molecular rotation. Only the intermolecular dipolar couplings are present. Even with this, the T_2' has not reached a plateau with the spinning speed. Therefore I feel pessimistic about the impact and outlooks applied to other more 'normal' molecules and solid samples.

In addition to the sentence already present in the conclusion section:

“The fast molecular dynamics present in camphor lead to a weakened dipolar network, which is in contrast to the refocused linewidths measured for rigid organic solids (e.g. tyrosine) that are currently still slightly larger (typically 50-200 Hz at 100 kHz MAS)¹⁵ than typical ¹H-¹H J-couplings.”

the following phrase was added to clarify that the application of the J-based experiments used here for camphor at 100 kHz MAS and above is not envisaged for rigid solid samples:

“and which prevents the approach here from being broadly applicable to rigid solids.”

The 2D correlation with J and D couplings turned out expected. There are still some visible features or 'artifacts' that are likely caused by residual homonuclear dipolar couplings.

We do not see any features or artifacts that are caused by residual homonuclear dipolar couplings in the 2D correlation spectra of Figure 3 (certainly not with an intensity comparable to the J-mediated cross-peaks). However, we note that the language used in the paragraph describing the folding artifacts that are common to all JRES spectra (liquids or solids) was potentially misleading, and we have now corrected it for clarity to:

“It should be noted that all the solid-state 2D JRES spectra acquired here contain weak folded artifacts in the ω_1 dimension (Figure S3), that are typical for any 2D JRES spectrum³ (for liquids or solids) and which are due to the use of an incomplete phase cycle and imperfect 180° pulses. (We used 4 steps instead of 16 to save time in these experiments which require very high digitization in t_1 .) Note that Figure S7 shows a spectrum acquired using a simpler pulse sequence that leads to broader mixed-phase lineshapes in F1, but where the artifacts are absent. The spectral width in ω_1 was therefore adjusted to minimize overlap with the center peaks. Nonetheless, partial overlap with the center peaks lead to slight asymmetry in the multiplet patterns (Figure S3).”

For camphor, various proton sites are well resolved. Do the shift differences impact the MAS average of the homonuclear dipolar couplings?

The effect of the chemical shift difference on the dipolar coupling under MAS has been previously extensively described in the literature. For example, by Simoes de Almeida et al. (<https://doi.org/10.1063/5.0055583>) or Duma et al. (<https://doi.org/10.1002/cphc.200301213>) for the case of spin-echoes. In general, shift differences will have an effect on averaging at higher order, but the only very special case would be when the chemical shifts were equivalent (see the next comment).

Have the authors tried the other 'well known' molecular crystal adamantane in which the CH and CH₂ sites have nearly identical shift. I would be curious on the effect of shift degeneracy on the MAS averaging of dipolar couplings.

This is an interesting point. Unfortunately, we do not expect to see any resolved J-splittings in adamantane because both peaks have a complex J-multiplet pattern, a triplet and a septuplet, small J-couplings (3-4 Hz), and very similar chemical shifts (separated by roughly 0.1 ppm, i.e. 90 Hz on a 900 MHz spectrometer), and this is confirmed in the JRES spectrum shown below.

Figure R1. JRES spectrum recorded at 100 kHz MAS on adamantane. As expected, there are no visible resolved splittings in the J-dimension.

The setting of magic-angle is very critical for observing small J-couplings and it will become even more for molecules with stronger ^1H homonuclear dipolar couplings. I would be keen to see how the J-res spectra change with the angle setting instead just making the T_2' longer stated for the angle calibration.

We have now included a series of JRES spectra recorded with different missets of the magic angle in supplementary Figure S9. As might be expected, magic-angle misset results in progressive broadening in F1 of the 2D JRES. It is important to note that the magic angle setting is not “ultra-sensitive”, in the sense that good JRES spectra are obtained at the magic angle found by standard angle optimization using T_2' , and that the degree of misset required to obscure the couplings is significant (i.e. to go from the best result at the 13.00 goniometer setting in figure S9 to for example the visibly degraded spectrum at 12.20 is well within the accuracy and sensitivity of the standard procedures for MAS setting).

Also, the low-contour plot for the 2D J-res in SI shows spinning sidebands in F1. Would that problem be solved by rotor-sync t_1 evolution time?

As discussed above, and now clarified in the text, the weak artifacts seen in the low level contour plot are not spinning sidebands, but they are due to the incomplete phase cycling of the 180° pulse. The spectra are in fact all rotor synchronised in t_1 . (The spectrum shown in figure S3 has a spectral width in F1 of 1428.57 Hz at a MAS rate of 168.157 Hz).

To support this, we now also show in Figure S7 a solid-state JRES spectrum of camphor obtained with a magnitude mode pulse sequence, with complete phase cycling, where these artifacts are not present (as expected), but where the lineshapes are mixed absorption/dispersion, (also as expected for magnitude mode).

Reviewer #3 (Remarks to the Author):

This study explores the observation and use of ^1H - ^1H J-couplings in plastic crystals using fast MAS solid-state NMR spectroscopy. The results are very interesting, especially the use of a cutting-edge 0.4 mm probe to enable the fastest spinning at around 170 kHz.

We thank the reviewer for their positive appreciation.

Nevertheless, I have some major concerns before consideration for publication.

1. Figure 1c, why the height of doublet is not equal, such as H_3' and H_3 ?
2. Left top of Page 4, please explain why there are “folded artifacts in the ω_1 dimension (Figure S3)”. This should be briefly mentioned in the main text instead of simply citing other works. This is very important, because it is related to the correct setting of experimental parameters.

The reviewer's points 1 and 2 are connected. The intensities of the two peaks forming the doublet of H_3' and H_3 are not equal due to the partial overlap with the folded artefacts that are common to JRES spectra. Such an overlap is clearly seen for H_3' in Figure S3, but it is less obvious for H_3 . As mentioned in the response to reviewer 2 above, the description of these folding artifacts has now been clarified to:

“It should be noted that all the solid-state 2D JRES spectra acquired here contain weak folded artifacts in the ω_1 dimension (Figure S3), that are typical for any 2D JRES spectrum³ (for liquids or solids) and which are due to the use of an incomplete phase cycle and imperfect 180° pulses. (We used 4 steps instead of 16 to save time in these experiments which require very high digitization in t_1 .) Note that Figure S7 shows a spectrum acquired using a simpler pulse sequence that leads to broader mixed-phase lineshapes in F1, but where the artifacts are absent. The spectral width in ω_1 was

therefore adjusted to minimize overlap with the center peaks. Nonetheless, partial overlap with the center peaks lead to slight asymmetry in the multiplet patterns (Figure S3).”

And the following sentence was added:

“Nonetheless, partial overlap with the center peaks can lead to slight asymmetry in the multiplet patterns. (Figure S3).”

3. In Table S3, the spectral width of F1 dimension for BABA-xy16 experiment is 9999.966 kHz? Please double check.

The value was corrected to 10000.

4. Figure 2c, there is an abnormal increase of linewidth at 150 kHz. Could the authors explain why?

The apparent increase in linewidth for H3 (and H3') at 150 kHz rate is within the fitting errors of ± 0.5 Hz. As mentioned in the main text, a prototype 0.4 mm MAS probe was used in this study on which the magic angle needed to be readjusted at each spinning rate, introducing some degree of error. The following comment was added to the main text:

“Notably, this explains the apparent increase in the refocused linewidths at 150 kHz MAS for H3' and H3, which is within the fitting error of ± 0.5 Hz.”

5. The authors mentioned that coherence lifetimes are longer than 20 ms at 160 kHz in the abstract. Please provide the experimental data about coherence lifetimes at different spinning rate.

Figure 2C has been updated to include the measured T2' values as a function of MAS rate.

6. Figure S7, it seems that the proton linewidths do not decrease with increasing spinning rate. Could the authors comment on this strange behavior?

In the 1D ^1H spectra of camphor, the linewidth is dominated by the inhomogeneous interactions, notably structural disorder and ABMS (as well as any B_0 field inhomogeneity). Altogether, these MAS independent contributions represent roughly 75% of the total ^1H 1D linewidth. Increasing the MAS rate therefore has only a very small effect on the total 1D ^1H linewidth, i.e. a decrease from 71 to 67 Hz for H3' and H3 (measured in TopSpin, without fitting to a doublet); and from 88 to 66 Hz for H8/9. A comment has been added to the caption of Figure S7.

7. Figure S6, it is a solution JRES spectrum for camphor dissolved in DMSO. Why does the spectrum look so noisy, especially at the chemical shift of 1.3ppm? Usually there should be a clear splitting for each peak in solution NMR spectrum.

We have updated the SI with a better solution JRES spectrum in Figure S6. The phase-sensitive 2D JRES spectrum was acquired using the Bruker pulse sequence jresgpph.

8. Usually the JRES experiment is amplitude modulated, but the authors used phase sensitive JRES experiment. Please explain the advantages of phase sensitive experiment and cite the relevant literatures about JRES experiment.

JRES experiments in solution typically use pulse sequences that employ field gradients to select coherence transfer pathways, and pseudo-pure absorption mode lineshapes are obtained by using a sine-bell window function and a magnitude representation. In solids probes, field gradients are not routinely available, so here pulse sequences using phase cycling and the States-TPPI method were used to obtain spectra with pure absorption lineshapes. (We note that for the purposes here, it does not really matter which pulse sequence is used, and it may well be that a more efficient acquisition scheme could be proposed in the future.) As mentioned above, we have now added a magnitude-mode 2D JRES spectrum of camphor in Figure S7.

9. Caption of Figure 2, please use “JRES” instead of “J-RES” for consistency through the main text.

Corrected.

10. Figure 3A, for the correlation between H3' and H4, why the peak is negative on the left but positive on the right of the diagonal peak? Similar negative peaks are observed in Figure 3B for the (H3', H4) cross peaks. Besides, why there are strong t1-noise at 1.0ppm in Figure 3A, but it is not observable in Figure 3B.

First, regarding t1-noise in the INADEQUATE spectrum, we thank the reviewer for highlighting this. On closer inspection we found that the increased t1-noise was caused by intermittent interfering external noise signals that occur over short time periods (~20 minutes), every few hours. (We are trying to track down the source of these 900 MHz signals!)

Fortunately, these interfering signals were strongest towards the end of the INADEQUATE acquisition, so we now show a spectrum with a t_1^{\max} of 43.6 ms (out of the original 102 ms), where the NMR signals have anyway almost completely decayed. **The resulting spectrum now shown in Figure 3 now has significantly reduced t1-noise.** (Luckily, the UC2QFCOSY did not have any of these spurious interference signals).

Second, regarding the negative peaks, these arise from the setting of the tau delays in the INADEQUATE and UC2QFCOSY experiments, and the fact that H3' and H4 are part of a larger spin system that includes H3 and H5'. Lineshapes in INADEQUATE type spectra in multiple spin systems have been discussed in detail, for example in Reference 37 (<https://doi.org/10.1016/j.jmr.2007.05.016>) The mixed in-phase / anti-phase character is perfectly in line with expectations. **We have now added a comment to the main text to highlight this.**

11. One of the key issues for observation of 1H-1H J couplings in solid-state NMR is to make sure that the proton dipolar couplings are completely suppressed. In fact, the strength of residual 1H-1H dipolar couplings could be on the same level as J couplings. So, how do the authors can guarantee that the dipolar couplings are completely removed, and they are actually observing the 1H-1H J coupling instead of residual 1H-1H dipolar coupling.

As discussed in the text, and now further clarified in the text in response to Reviewer 1 above, the use of plastic crystals here actually allows us to very clearly distinguish between J and dipolar coupling effects.

Particularly, the authors mentioned that the "magic angle needed to be readjusted at each spinning rate".

Insufficiently accurate magic angle may also introduce additional residual dipolar couplings. Careful discussion about this point is very important, since we can see that the obtained J(H3'-H3) is larger than that obtained from solution NMR.

The adjustment of the magic angle is not because the experiment is unusually sensitive to the angle, but because we used a prototype probe where the angle changes slightly when the spinning rate is changed, due a change in probe temperature when the gas flows are different!

To support this, we have now included a series of JRES spectra recorded with different missets of the magic angle in supplementary Figure S9. As might be expected, magic-angle misset results in progressive broadening in F1 of the 2D JRES. It is important to note that the magic angle setting is not "ultra-sensitive", in the sense that good JRES spectra are obtained at the magic angle found by standard angle optimization using T_2' , and that the degree of misset required to obscure the couplings is significant (i.e. to go from the best result at the 13.00 goniometer setting in figure S9 to for example the visibly degraded spectrum at 12.20 is well within the accuracy and sensitivity of the standard procedures for MAS setting).

12. The authors mentioned in the main text "a detailed analysis shows that in addition to the residual dipolar broadening, the one-dimensional linewidth contains contributions from anisotropic bulk magnetic susceptibility (ABMS), and structural disorder. Both of these latter contributions are inhomogeneous, and thus refocusable." The statement may not be accurate, since I believe the higher-order ABMS component is actually not refocusable.

We have modified the sentence to read: “Both of these latter contributions are inhomogeneous (excluding the effects of potential higher order cross terms), and thus refocusable.”

Similarly, the inherent structural disorder induced chemical shift dispersion can be refocusable?

Yes, the inherent structural disorder induced chemical shift dispersion is refocusable. To clarify, we have modified the phrase to read: “the one-dimensional linewidth contains contributions from anisotropic bulk magnetic susceptibility (ABMS), and inherent structural disorder”

13. In the main text, the authors should summarize all the obtained J values by fitting the columns in Figure 1c into a Table in order to compare the experimental results to that of solution NMR.

The only coupling that can be accurately measured is $J_{H3-H3'}$, and this value is compared to solution in the text. The other couplings are less well resolved, and lead to more complex multiplet structures, and fitting the lineshapes leads to overfitting. We do show in Figure S2 a comparison of lineshapes between the experimental solid-state JRES columns for all the peaks with the predicted lineshapes if the couplings were the same as in solution.

14. Please provide important experimental parameters in the Figure captions. For example, for Figure 3, the recoupling time for each experiment should be provided.

Done.

REVIEWER COMMENTS

Reviewer #3 (Remarks to the Author):

I appreciate the authors' great effort in improving the quality of the manuscript.

We thank the reviewer for their positive evaluation.

Before the acceptance, I still have some minor comments.

1. Regarding to the "weak folded artifacts in the ω_1 dimension" in Figure 3a (or Figure S3), can the authors comment on how does the folding happen in the ω_1 dimension and is there any approach for overcoming this problem?

The current text in the first revised manuscript had already been updated to address this question, as originally posed by reviewers 2 and 3, to read:

"It should be noted that all the solid-state 2D JRES spectra acquired here contain weak folded artifacts in the ω_1 dimension (Figure S3), that are typical for any 2D JRES spectrum³ (for liquids or solids) and which are due to the use of an incomplete phase cycle and imperfect 180° pulses. (We used 4 steps instead of 16 to save time in these experiments which require very high digitization in t_1 .) Note that Figure S7 shows a spectrum acquired using a simpler pulse sequence that leads to broader mixed-phase lineshapes in F1, but where the artifacts are absent."

The first revised manuscript thus already included a spectrum that was acquired without the folded artifacts (Figure S7).

We have now slightly reworded this paragraph to make sure there is no future confusion:

"It should be noted that all the solid-state 2D JRES spectra acquired here contain weak folded artifacts in the ω_1 dimension (Figure S3), that can be found in any 2D JRES spectrum³ (for liquids or solids) and which are due here to imperfect 180° pulses in combination with the use of an incomplete phase cycle. (We used 4 steps instead of 16 to save time in these experiments which require very high digitization in t_1 .) These artifacts can be removed, as shown in Figure S7 with a spectrum acquired using a simpler pulse sequence that leads to broader mixed-phase lineshapes in F1, but where the artifacts are absent."

2. In Table S3, the author corrected "the spectral width of F1 dimension for BABA-xy16 experiment" to be 10000 kHz, i.e. 10 MHz. Why using such a huge spectral width? Such spectral width is also completely not rotor-synchronized. Why?

We thank the reviewer for picking this up. This was typo, which has now been corrected to 10 kHz (indeed, not MHz!).

3. In the revised SI, somehow Table S4 is missing, but TableS1-S3 and Table S6-S8 are there.

We thank the reviewer for picking this up. The SI table numbering has been corrected.